# ZERO-SHOT VISUAL CLASSIFICATION WITH GUIDED CROPPING

## ABSTRACT

Pretrained vision-language models, e.g., CLIP, show promising zero-shot transfer capability across various unseen classification datasets. However, there is an inherent limitation: CLIP image encoders are typically designed to extract generic image-level features that summarize superfluous or confounding information for the target tasks. This results in degradation of classification performance, especially when objects of interest cover small areas of input images. In this work, we propose CLIP with Guided Cropping (GC-CLIP), where we use an off-the-shelf zero-shot object detection model in a preprocessing step to increase the focus of zero-shot classifiers on the object of interest and minimize the influence of extraneous image regions. We empirically show that our approach improves zero-shot performance across architectures and datasets, most favorably for small objects.

## 1 INTRODUCTION

Conventional supervised learning for classification tasks involves training deep neural networks on labelled datasets (He, 2020). The resulting models are inherently limited by the class definitions of a specific task. In contrast, recent research focuses on open-vocabulary classification models (Jia et al., 2021; Radford et al., 2021). Pretrained with large-scale image-text datasets, these models define target classes generically through natural language, and generally have zero-shot transfer capability, being able to perform on any unseen classification datasets without further training.

CLIP (Radford et al., 2021) is one of the most popular open-vocabulary classifiers. Its architecture comprises image and text encoders which encode input images and texts into a shared latent space. These encoders are trained with a contrastive loss such that the dot product similarity between image and text encodings indicate how likely input images and texts correspond to one another.

A CLIP's limitation lies in the fact that its encoders are designed to be generic in the sense that its image encodings encompass entire information of a given image regardless of the target task. While this behavior is desirable for some problems, it simultaneously poses a limitation when performing classification on unseen datasets where only certain labels and image contents are of interest. In these cases, encoding entire image contents can lead to suboptimal performance, particularly for small objects. E.g., in Figure 1a, the large water region in the image dominates similarity scores between image and text encodings of water-related classes, leading to an incorrect zero-shot prediction.

Our central question is: How can we reduce non-discriminative and extraneous information from the image encodings? We observe that reducing areas of context regions by cropping input images around objects of interest can be beneficial. Figure 1b illustrates that the cropped image with reduced water regions decreases similarity scores of incorrect water-related classes and results in the dominant similarity score of the correct class (i.e., canoe).

One approach to reduce influence from non-discriminative information is to explicitly crop extraneous regions. One possibility is to employ open-vocabulary object detection models directly for classification. These models produce object bounding boxes and *locally* categorize them based on any given text prompts (Minderer et al., 2022; Kuo et al., 2022). We show, however, that these approaches are in themselves not optimal for image classification tasks. We conduct an experiment to extend one of the most recent open-vocabulary object detection models OWL-ViT (Minderer et al., 2022) for classification, where each sample belongs to only one class. We observe that, while

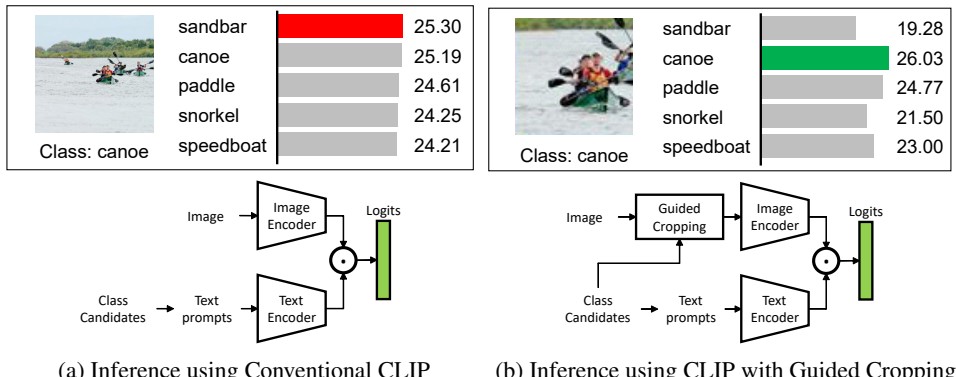

Figure 1: Logits from CLIP (ViT-B/32) before and after cropping around objects of interest

OWL-ViT provides reasonable bounding box estimation, its zero-shot classification performance is inferior to CLIP baselines (more details in section 5.5).

In this work, we aim to improve the zero-shot object classification performance of CLIP by guiding its focus to objects of interest and reducing influence of unrelated visual information. Instead of using OWL-ViT for classification directly, we propose to employ it as a pre-processing bounding box extraction module, such that cropped input images are processed by CLIP (Figure 1b). We refer to this approach as CLIP with Guided Cropping (GC-CLIP). We show that classification performance greatly depends on the choice of cropping scales, particularly for images with small objects.

Our contributions are as follows: We provide evidence that generic CLIP encoders can lead to suboptimal zero-shot transfer performance, particularly on the images with small objects. We propose a method to improve zero-shot CLIP using bounding boxes estimated from a state-of-the-art open-vocabulary object detector. We conduct experiments to show that our approach outperforms a classifier built directly from this detector, as well as other baselines across different scenarios. Lastly, we conduct ablation studies analyzing the conditions under which our approach works well.

## 2 RELATED WORK

**Zero-Shot Learning and Zero-Shot Transfer**    In conventional zero-shot learning, models recognize images of unseen classes based on their known semantics (Akata et al., 2015; Li et al., 2021; Naeem et al., 2021; Mancini et al., 2021). In this work, we focus on zero-shot transfer and aim to evaluate model performance on unseen datasets - classes in those datasets may not be completely unseen to the model, however images of target datasets are unseen.

**Open-Vocabulary Classification**    Open-vocabulary classification models enable zero-shot transfer by using natural language to define class semantics, affording greater flexibility in the task definition without requiring expensive annotations. Images and text prompts can be projected by image/text encoders into a joint embedding space so that their similarities can be computed. CLIP (Radford et al., 2021) and ALIGN (Jia et al., 2021) encourage similarity between image-text pairs based on contrastive losses. Menon & Vondrick (2022) improves zero-shot performance by using multiple text prompts per category based on queries from large language models. Florence (Yuan et al., 2021) considers more modalities in addition to images and texts.

While these models perform well in open-world scenarios, their performance can be limited for certain inputs as their encoders may encode extraneous information. CALIP (Guo et al., 2023) looks for discriminative information by incorporating attention information in feature-level. This relies on the quality of CLIP attention maps which can be poor in many cases (Chen et al., 2022). On contrary, we seek discriminative information directly at an image-level, which is more interpretable.

**Open-Vocabulary Object Detection**    Open-vocabulary object detectors produce bounding boxes given input text prompts (Gu et al., 2021; Zhong et al., 2022; Li et al., 2022; Kuo et al., 2022; Zhang et al., 2022). ViLD (Gu et al., 2021) trains an object detector based on knowledge distillation

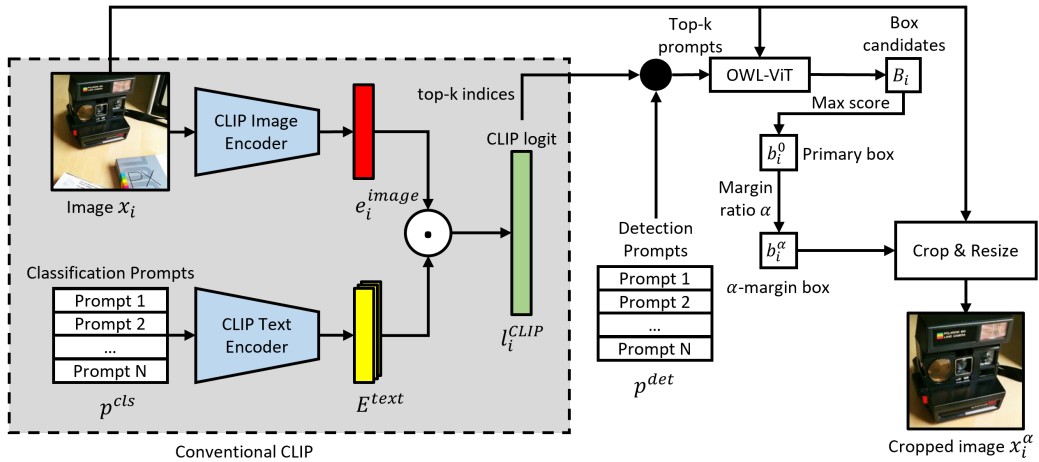

Figure 2: Guided Cropping pipeline to obtain a guided cropped image with margin ratio $\alpha$

from pretrained open-vocabulary classification models. In OWL-ViT (Minderer et al., 2022), simple modifications of standard vision transformers are fine-tuned with large-scale image-text datasets for object detection. GLIPv2 (Zhang et al., 2022) extends models to handle various localization tasks.

Object detection models have innate ability to not only localize, but classify localized objects based on local information. A question may be raised, whether they are in general sufficient to solve the zero-shot classification task alone. In section 5.5, we conduct experiments based on OWL-ViT, a recent off-the-shelf model, and demonstrate its poor performance on classification tasks. In this work, we use the open-vocabulary object detection models only for bounding box extraction.

## 3 BACKGROUND

**Problem Formulation** Given a test dataset $\{(x_i, y_i)\}_{i=1}^{N_s}$, where $x_i \in \mathcal{X} = \mathcal{R}^{w \times w}$ and $y_i \in \mathcal{Y} = \{1, 2, \ldots, N_c\}$ is an image and its corresponding label, our task is to construct a prediction function $F : \mathcal{X} \to \mathcal{Y}$ based on pretrained open-vocabulary models to maximize $P(\hat{y}|x) = P(F(x)|x)$ without accessing any test samples. The remainder of this section describes such a prediction function based on CLIP, and our approach will be presented in section 4.

**Conventional CLIP** CLIP (Radford et al., 2021) is a multi-modal model with zero-shot transfer capability. It consists of an image encoder $G$ and a text encoder $H$. To perform classification on an unseen target dataset, a text prompt $p_j^{cls}$ needs to be defined for each target class $j \in \mathcal{Y}$. Then, an embedding of each prompt can be obtained by: $e_j^{text} = H(p_j^{cls})$. During inference, an input image $x_i$ will be projected into its image embedding $e_i^{image} = G(x_i)$ so that its classification logit $l_i^{CLIP}$ can be computed as:

$$l_i^{CLIP} = (E^{text})^T e_i^{image} = \begin{bmatrix} e_1^{text} & e_2^{text} & \ldots & e_{N_c}^{text} \end{bmatrix}^T e_i^{image}. \tag{1}$$

Each entry $l_{ij}^{CLIP}$ of the logit indicates the similarity score between the (embedded) input image and the $j$-th prompt. The final class prediction can then be obtained as $\hat{y}_i = \arg\max_{j \in \mathcal{Y}} l_{ij}^{CLIP}$. Here, we assume that one prompt is available per class. However, (Menon & Vondrick, 2022) has recently shown that multiple prompts per class can improve performance. In this case, $e_j^{text}$ from equation 1 can be replaced with the average embedding computed from all available text prompts of class $j$.

## 4 METHODOLOGY

### 4.1 CLIP WITH GUIDED CROPPING

Conventionally, an image embedding $e_i^{image}$ is computed directly from the full image $x_i$ without any task-specific constraints. This implies that potentially unrelated information is also encoded

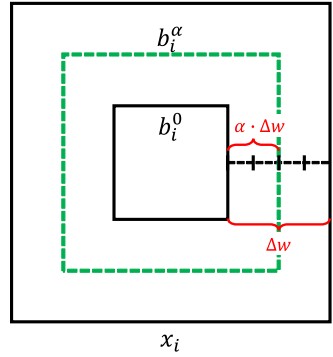
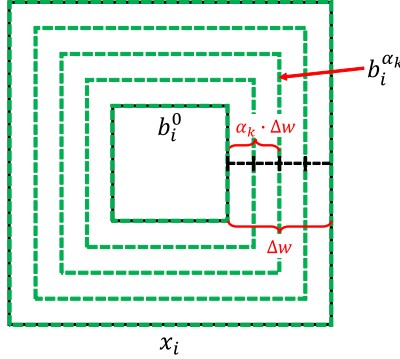

(a) Without augmentation

(b) With Multi-Margin augmentation

Figure 3: Each green square corresponds to a final bounding box $b^\alpha$ (or $b^{\alpha_k}$) which will be used to crop the original image $x_i$ to produce logit for the final prediction. $\Delta w$ is the width difference between the original image and the primary box $b_i^0$. $\alpha$ and $\alpha_k$ are margin ratios.

into $e_i^{image}$, especially in cases of a small object image, which may lead to suboptimal performance (see section 5.3). Minimizing the amount of unrelated concept information in image embeddings is desirable in this case. Our approach GC-CLIP achieves this by using bounding box estimates from a Guided Cropping component.

For our Guided Cropping, in theory, any detectors which can localize target objects without further supervision can be employed. Our goal in this paper is to show that, there is at least one detector which, under our framework, can improve overall performance of CLIP. In our work, we choose OWL-ViT (Minderer et al., 2022), the state-of-the art open-vocabulary object detector as a candidate.

OWL-ViT takes an image and text prompts of target classes as inputs and produces outputs as a set of bounding boxes together with their scores and classes. In this work, we only use OWL-ViT as a bounding box extraction module as its class predictions are not accurate enough (see section 5.5). The overall GC-CLIP pipeline is shown in Figure 2. We only consider top-k classes (we use k=5) to refine the preliminary CLIP predictions. This is reasonable since it has high probabilities that these top-k classes contain the correct class (see appendix A.4).

**Candidate box extraction**   We detect bounding boxes of each top-k class with OWL-ViT independently. We found that this is more robust to misdetection resulting in better performance compared to detecting bounding boxes of all classes at once (see appendix A.6). Formally, a set of bounding box candidates $B_i$ for an image $x_i$ can be obtained based on OWL-ViT as follows:

$$B_i = \bigcup_{j \in J_i^k} b_{ij} = \bigcup_{j \in J_i^k} OWL(x_i, p_j^{det}) \tag{2}$$

where $J_k \subseteq \mathcal{Y}$ is a set of top-k classes with respect to $l_i^{CLIP}$, $p_j^{det}$ is a text prompt for detection of class $j$ and $OWL$ is OWL-ViT detection function returning a max-score bounding box with respect to an input image and a prompt. All bounding boxes are adjusted to squares to avoid skewing images when they are, afterward, transformed into a CLIP-compatible image size. (e.g., $224 \times 224$).

**Box selection**   Next, we need to pick one bounding box from $B_i$. We start from a primary box $b_i^0 \in B_i$ which has the highest estimated score from OWL-ViT. In our experiments, we found that using the primary box directly is generally suboptimal as its crop may be too tight. It is therefore beneficial to slightly enlarge the box (see section 5.2). Given $b_i^0$ has the width of $w_{b_i^0}$ and $x_i$ has the width of $w$, the box is enlarged to an $\alpha$-margin box $b_i^\alpha$ uniformly in all direction to the size of $w_{b_i^0} + \alpha(w - w_{b_i^0})$, where $\alpha \in [0, 1]$ is called the margin ratio (see Figure 3a). For the enlargement, if a box edge exceeds image boundary in one direction, the enlargement will be compensated in the opposite direction. In cases with box augmentation, multiple $\alpha$ can be employed (see section 4.2).

**Logit computation**   This selected box $b_i^\alpha$ is used to crop $x_i$ and resize it to a CLIP-compatible image size $w \times w$ resulting in a preprocessed image $x_i^\alpha$. The new top-k logit $l_i^{GC\_CLIP(k)}$ is computed

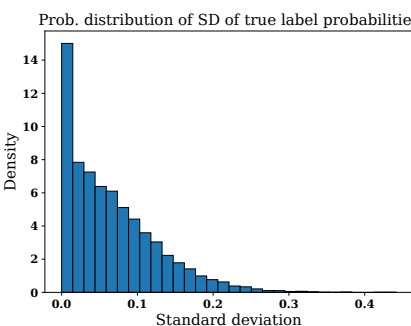
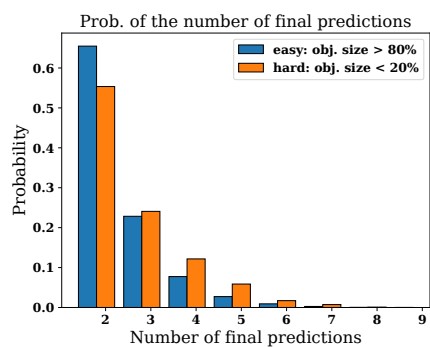

(a) SD of predicted true-label probabilities     (b) Number of final class predictions across crops

Figure 4: Results when forwarding multiple random crops of the same images (from ImageNetS919 dataset) to CLIP (ViT-B/32) demonstrating CLIP sensitivity to non-semantic changes.

based on $x_i^\alpha$ as follows:

$$l_i^{GC\_CLIP(k)} = \begin{bmatrix} e_{j^1}^{text} & e_{j^2}^{text} & \cdots & e_{j^k}^{text} \end{bmatrix}^T G(x_i^\alpha), \tag{3}$$

where $j^1, j^2, \ldots, j^k \in J_i^k$. The final class prediction is the class within $J_i^k$ corresponding to the maximum entry of $l_i^{GC\_CLIP(k)}$.

## 4.2 TEST-TIME BOX AUGMENTATION

Working with raw/preprocessed input images directly can lead to noisy prediction from CLIP. Small non-semantic changes in images can cause changes in predictions making CLIP outputs difficult to analyze. We show this behavior by processing 10 random crops (90%-100% of the original widths) of the same image with CLIP. One would expect standard deviations of its predicted true-label probabilities to be low and its final class predictions not to change across different crops. However, we notice from Figure 4a that the standard deviations can be relatively high (around 0.2), while the average true-label probability is 0.55. In addition, only around 60% of test samples have no changes in final class predictions across crops (see Figure 4b). It is also observable that samples with smaller object sizes have less reliable predictions. These results indicate that CLIP is quite sensitive to non-semantic changes. Therefore, instead of computing logits from raw/preprocessed images only, we can perform a simple test-time augmentation to help mitigate this issue. In the following, we investigate two augmentation strategies.

**Random Crop Box Augmentation (RAug)** With RAug, we augment a single input (raw or pre-processed) image into $N_{aug}$ total images by cropping the input image with $N_{aug}$ boxes of random widths within $[\beta w, w]$, while $\beta \in (0, 1)$. The augmented images are used to compute multiple predicted logits as per equation 3, which can then be averaged to produce the final logit score.

**Multi-Margin Box Augmentation (MAug)** In some cases, it is beneficial to consider context information as long as it does not dominate the object in question (Hoyer et al., 2019). With MAug, we need to firstly obtain the primary box $b_i^0$. Then, instead of using a margin ratio $\alpha$ as in section 4.1, we perform an object-centric augmentation by using $N_{aug}$ bounding boxes obtained from multiple margin ratios, distributed uniformly from 0 to 1 (see Figure 3b). In other words, the set of all final boxes used in this augmentation is $\left\{ b_i^{\alpha_k} | \alpha_k = \frac{k}{N_{aug}-1}, k \in \{0, 1, \ldots, N_{aug} - 1\} \right\}$. Similarly, logits computed from images cropped by these final boxes are then averaged to get the final logit score.

It must be noted that, with MAug, regions close to the target object are covered by more boxes compared to regions far from the object. Therefore, this augmentation strategy allows some context information to be considered but with lower importance compared to the object's immediate context.

## 5 EXPERIMENTS

In this section, we conduct experiments to demonstrate that utilizing CLIP with Guided Cropping can improve zero-shot transfer performance of CLIP. In addition, several ablation studies are conducted to understand the failure modes and the conditions under which our approach works well.

**Datasets:** We aim to show that generic CLIP encoders can lead to suboptimal zero-shot performance, particularly on the images with small objects. We showcase the effectiveness of our GC-CLIP in such cases. Therefore, we study datasets in which object sizes in images are controllable. We find two datasets - ImageNetS919 (Gao et al., 2022) and CUB (Welinder et al., 2010) fit this criteria. These datasets provide segmentation/bounding box annotations from which object sizes of image samples can be obtained and enable us to quantify the performance on objects covering small areas. Details of these two datasets are as follows - (1) ImageNetS is an extension of ImageNet and originally designed for unsupervised semantic segmentation. We use validation split of the dataset in which pixel-wise segmentation annotations are available. It contains 12,419 samples of 919 classes in total. We construct a subset with target objects of small sizes, referred as ImageNetS919-SM, containing 2,334 samples whose object sizes are no more than 20% of the full image size. (2) CUB is a benchmark for fine-grained classification consisting of 200 bird types. We evaluate our models on its test split of 5,794 samples. Based on bounding box annotations, we construct a subset whose target object sizes are less than 20% of the full image size resulting in CUB-SM containing 1,390 samples. Details of our dataset splitting methodology can be found in appendix A.1.

**Baselines:** We employ CLIP (Radford et al., 2021) variations as well as CALIP (Guo et al., 2023) as our baselines. DataComp represents a recent variation of CLIP from (Gadre et al., 2023). Two classification prompt types are investigated (1) Category: Each class has a single prompt of its category name (2) Descriptions: Each class has multiple prompts queried automatically from GPT-3 according to Menon & Vondrick (2022). In the latter case, the final logit value for a given class is computed by averaging the logit values obtained from all prompts for that class.

**Implementation:** We apply our Guided Cropping and box augmentation on top of each baseline. For Guided Cropping variations, the margin ratio $\alpha$ of 0.2 is used unless otherwise specified. We perform box augmentation with $N_{aug} = 11$. For RAug, $\beta = 0.9$ is used. The high value of $\beta$ makes RAug augmented boxes less likely to crop object contents away. Different CLIP backbones like ViT-B/32, ViT-B/16 and ViT-L/14 are studied in this work. For OWL-ViT, its backbone is ViT-B/32 for all experiments. Category names are used as prompts to perform detection with OWL-ViT. The code of our implementation will be publicly available upon paper acceptance.

### 5.1 ZERO-SHOT TRANSFER RESULTS

We evaluate zero-shot performance of different configurations on various datasets including both unconstrained object sizes (full dataset) and small-object variants (with -SM suffix). Results for ViT-B/32 and ViT-B/16 backends are shown in Table 1 (ViT-L/14 and DataComp in appendix A.2).

Considering datasets with unconstrained object sizes, ImageNetS919 and CUB, our Guided Cropping performance is comparable to (or slightly better than) non-Guided Cropping baselines. This is expected since many samples in these cases could have objects whose sizes already dominate the scene. On the other hand, both box augmentations consistently improve classification performance in all cases indicating that raw predictions from CLIP models are indeed noisy. Smoothing their predictions with box augmentations complement our methods to be more robust to this noise.

GC-CLIP demonstrates consistent improvement over baselines on datasets with small objects (ImageNetS919-SM, CUB-SM) across different model/prompt configurations. This indicates that our approach, as expected, is more beneficial for images with small objects. This is reasonable since images with small objects leave more space in the images for context information which should be excised before performing image encoding. Another interesting observation is that employing MAug generally achieves better performance. This infers that hinting context cues with lower importance can indeed complement the focus on target objects to make definite and correct decisions.

In Table 2, we conduct an experiment with CALIP. Some observations can be seen from the results. Firstly, compared to Table 1, CLIP with Guided Cropping performance on ImageNetS919-SM and CUB-SM (55.18, 51.44) is better than CALIP performance (53.81, 50.36) even without box augmen-

Table 1: Zero-shot classification accuracies from different datasets and model configurations.

| Model | Prompt | Guided Cropping | Box Aug. | Dataset | | | |
|---|---|---|---|---|---|---|---|
| | | | | ImageNetS919 | CUB | ImageNetS919-SM | CUB-SM |
| CLIP (ViT-B/32) | Category | - | - | 63.62 | 51.83 | 52.83 | 49.57 |
| | | - | Random Crop | 64.42 | 52.45 | 53.47 | 50.79 |
| | | ✓ | - | 63.61 | 52.40 | 55.18 | 51.44 |
| | | ✓ | Random Crop | 64.46 | **53.12** | **56.00** | 52.81 |
| | | ✓ | Multi-Margin | **64.66** | **53.12** | **56.00** | **53.09** |
| | Descriptions | - | - | 68.54 | 53.05 | 55.70 | 50.14 |
| | | - | Random Crop | 69.15 | 53.62 | 57.33 | 50.79 |
| | | ✓ | - | 68.59 | 54.07 | 58.61 | **53.38** |
| | | ✓ | Random Crop | 69.07 | 54.47 | 59.08 | 53.09 |
| | | ✓ | Multi-Margin | **69.62** | **54.56** | **60.07** | 52.95 |
| CLIP (ViT-B/16) | Category | - | - | 68.60 | 56.51 | 57.75 | 55.54 |
| | | - | Random Crop | 68.81 | 56.89 | 58.05 | 57.41 |
| | | ✓ | - | 68.06 | 56.09 | 58.65 | 55.97 |
| | | ✓ | Random Crop | 68.19 | 56.78 | 58.35 | 57.12 |
| | | ✓ | Multi-Margin | **68.94** | **57.30** | **59.81** | **57.63** |
| | Descriptions | - | - | 72.67 | 57.78 | 61.61 | 56.55 |
| | | - | Random Crop | 73.17 | 58.87 | 62.13 | 57.99 |
| | | ✓ | - | 72.61 | 58.70 | 63.28 | **59.35** |
| | | ✓ | Random Crop | 72.86 | 58.99 | 63.32 | 58.78 |
| | | ✓ | Multi-Margin | **73.49** | **59.34** | **64.05** | 59.06 |

Table 2: Performance of CALIP with/without Guided Cropping using category-based prompts.

| Model | Guided Cropping | Box Aug. | Dataset | |
|---|---|---|---|---|
| | | | ImageNetS919-SM | CUB-SM |
| CALIP (ViT-B/32) | - | - | 53.81 | 50.36 |
| | - | Random Crop | 54.97 | 52.88 |
| | ✓ | - | 55.66 | 52.59 |
| | ✓ | Random Crop | **56.08** | **54.03** |

tation. Secondly, CALIP can be integrated with Guided Cropping to further improve performance. This demonstrates flexibility of our approach for combining with other classifiers.

A question may arise: how does Guided Cropping affect supervised models? We conduct experiments integrating our Guided Cropping with supervised models (see appendix A.3). For few-shot models, this integration can improve performance. Fully-supervised models benefit less from cropping. This is expected since these models tend to be more vulnerable to dataset biases. E.g., unrelated contexts could be used as shortcuts (Geirhos et al., 2020) to gain in-distribution performance.

## 5.2 IMPORTANCE OF MARGIN RATIO

Margin ratio ($\alpha$) mentioned in section 4.1 controls how much primary boxes are enlarged before they are used to crop input images. Varying margin ratios can help us understand how CLIP reacts to Guided Cropping from $\alpha = 0.0$ (crop with a raw OWL-ViT box) to $\alpha = 1.0$ (no Guided Cropping at all). We conduct an experiment with different $\alpha$ as shown in Figure 5. We mainly discuss results from GC-CLIP and GC-CLIP+RAug here as these configurations utilize a single $\alpha$.

According to the results, when Guided Cropping is applied ($\alpha < 1$), classification accuracies are generally better than the accuracies without Guided Cropping ($\alpha = 1$). This confirms the benefit of GC-CLIP. It must be noted that there are some consistent drops of the performance when the values of $\alpha$ are too small (e.g., when $\alpha \in [0.0, 0.1]$). Bounding boxes that are too tight can degrade classification performance. One explanation of this observation is that in order to recognize an object, models need to know the object shape clearly. Too tight bounding boxes can make the models have unclear information on the object boundaries leading to performance drops.

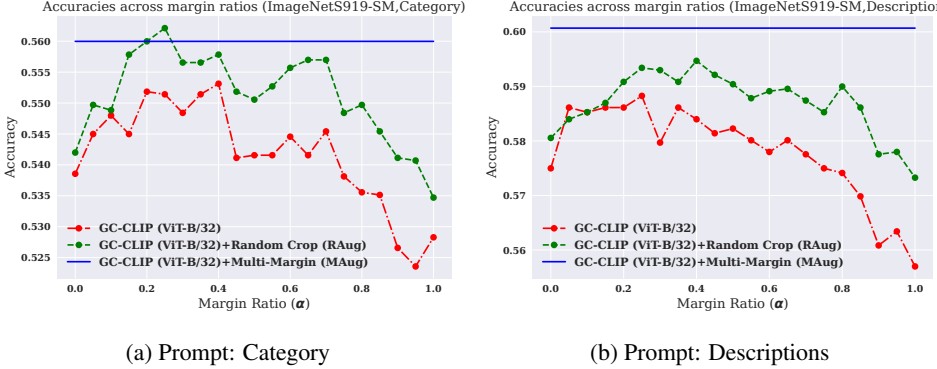

(a) Prompt: Category          (b) Prompt: Descriptions

Figure 5: Zero-shot accuracies on ImageNetS919-SM evaluated with different margin ratios.

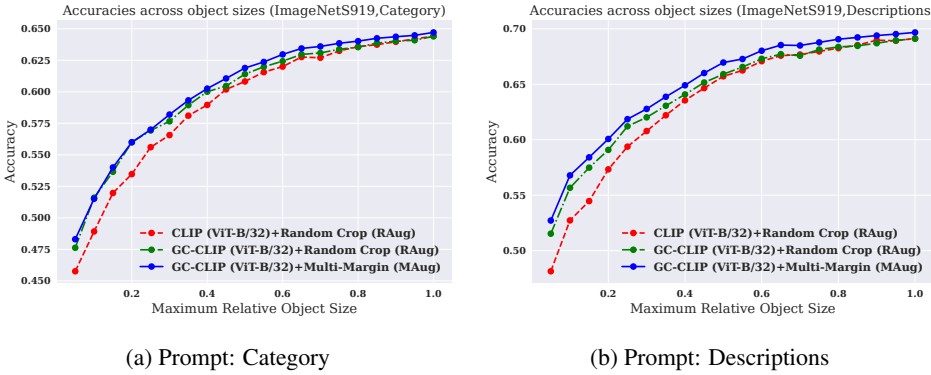

(a) Prompt: Category          (b) Prompt: Descriptions

Figure 6: Accuracies (ViT-B/32) on subsets of ImageNetS919 with various object size conditions.

## 5.3 UNDERSTANDING OBJECT SIZE CONDITIONS

Above we conduct experiments on small object images with one size condition (i.e., object size $< 20\%$ of image size). Here, we explore our approach on different object size conditions. We vary maximum relative object size of ImageNetS919 from 5% to 100% for our evaluation. The results are in Figure 6 (and appendix A.5). When object sizes are not constrained (i.e., x-axis = 1.0), Guided Cropping remains comparable to baselines (similar observation in Table 1). However, as maximum object sizes decrease, the accuracy gaps between conventional CLIP and GC-CLIP become larger. The gaps are also more significant when MAug is applied for box augmentation instead of RAug. This experiment highlights that our approach works well for images with small objects.

## 5.4 QUALITATIVE EVALUATION

We quantitatively evaluate GC-CLIP by visualizing some samples that are predicted differently than standard CLIP. Corrected samples are in Figure 7a. In the *container ship* image, "land" and "sea" are contexts spanning large image regions making standard CLIP falsely predict the input as *amphibious vehicle*. However, GC-CLIP categorizes the image by focusing on primary box at the watercraft.

On the other hand, samples whose predictions are incorrectly changed by GC-CLIP are in Figure 7b. These failures are due potentially to the distances between target objects and important contexts. While MAug allows some contexts to be considered, large distances between target objects reduce importance of the contexts for GC-CLIP (less boxes cover the contexts). E.g., considering the *space shuttle* image, the target object is so small that lacking any additional context, it is quite difficult to distinguish between a *missile* and a *space shuttle* (which is usually launched orthogonal to the ground). However, large distance between the ground and the object box reduces effects from the ground in GC-CLIP. Strategies to weight contexts dynamically can be investigated in future works.

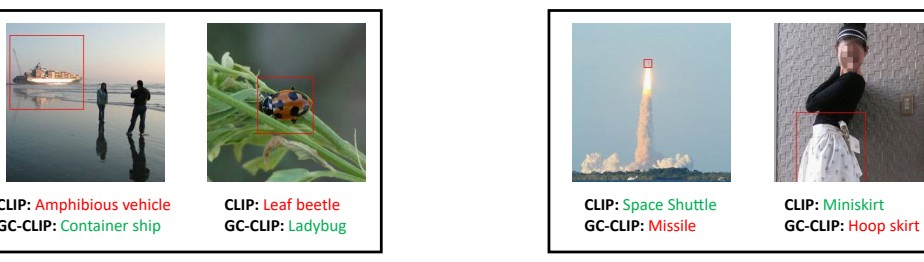

(a) Improved cases                                   (b) Failure cases

Figure 7: Predictions of CLIP (with RAug) and GC-CLIP (with MAug) with ViT-B/32 on ImageNetS919 samples. Red boxes represent primary boxes $b^0$ estimated from our GC-CLIP.

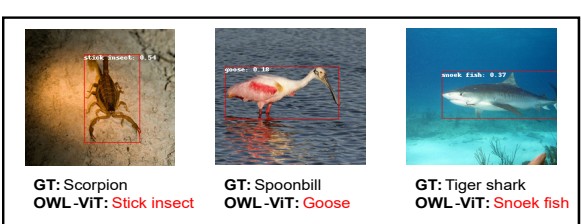

Figure 8: Examples of failure modes of the OWL-ViT based classifier.

## 5.5 Performance of OWL-ViT directly as a classifier

Here, we show that OWL-ViT, when adopted as a classifier directly, has subpar performance. In this case, we need to transform its outputs from sets of bounding box locations, scores and class labels into class-wise logits. Given an input image, the prediction logit of a class can be obtained as follows. We first iterate whether there are any bounding boxes exist for that class. If any exist, the class logit value is assigned as the maximum score among its boxes. Otherwise, its logit is zero. This simple extension encourages classes of bounding boxes with high scores to have high logits.

This classifier obtains 20.34% and 40.78% as top-1 and top-10 ImageNetS919 accuracies respectively which are low relative to baseline performance in Table 1. Figure 8 shows that OWL-ViT gives reasonable bounding boxes, but its class predictions are inaccurate and often confused with other semantically similar classes (e.g. tiger shark as a snoek fish). These results confirm that OWL-ViT is not optimal to be used as a classifier on standard classification benchmarks.

We hypothesize that this behavior might be attributed to the multi-task nature of the model. OWL-ViT utilizes a single image encoder to extract features that are used for both bounding box prediction and classification. Due to the limited capacity of the encoder or the choice of training strategies, it may compromise performance of individual tasks so that the average performance across tasks are reasonable but the performance of individual tasks may not be maximized.

## 6 Conclusion

We identify a clear limitation of CLIP-based models for zero-shot transfer on unseen classification datasets: as its image encoder is designed for encoding a generic image-level representation, it is prone to encode non-discriminative context information into image features leading to performance degradation, particularly for small objects. We propose GC-CLIP to reduce the effects from potentially non-discriminative information based on object bounding boxes estimated from an open-vocabulary object detection model. We empirically demonstrate that our approach outperforms baselines especially in cases of image samples with small objects. We analyze conditions in which our approach performs well in several additional ablation studies. We hope this work sheds a new light on the behavior of large-scale open-vocabulary models for classification and motivates future research to address this limitation in a more systematic manner.

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

# A APPENDIX

## A.1 CONSTRUCTING DATASET VARIATIONS WITH SMALL OBJECTS

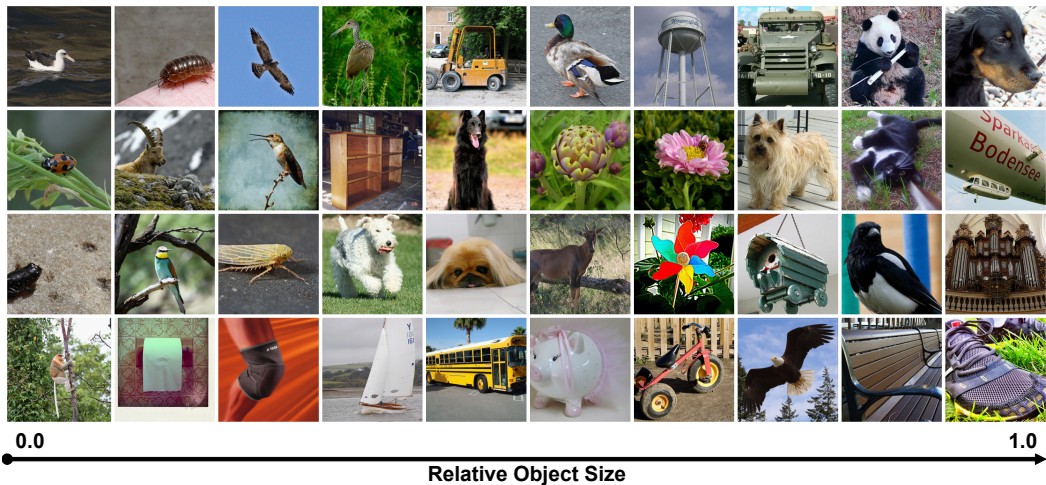

Figure 9: Example images from ImageNetS919 with different relative object sizes.

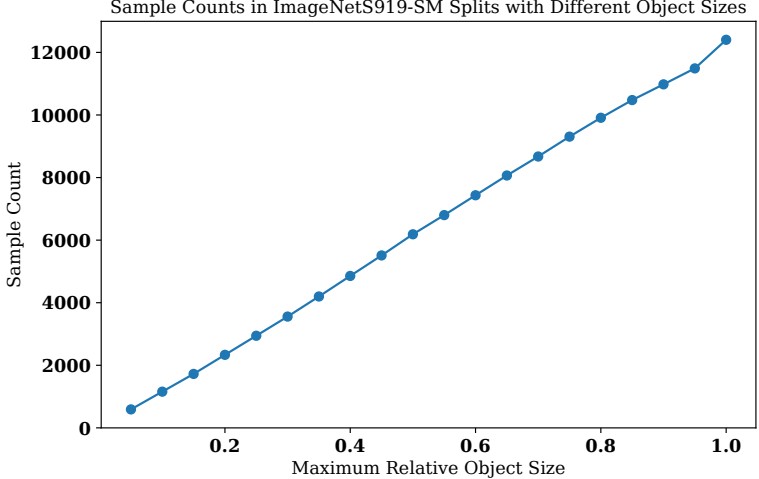

Figure 10: The number of samples in each object size condition of ImageNetS919.

In section 5, we use datasets based on ImageNetS and CUB as well as their small object variations (e.g., ImageNetS-SM and CUB-SM). In this section, we provide more details how those small variations are constructed.

For each image sample, its object size is computed based on object bounding box. In case of CUB, the bounding box is obtained directly from available annotations. However, for ImageNetS, only its pixel-wise segmentation is provided. In this case, object bounding box can be extracted from the segmentation in terms of minimum and maximum coordinates along $X$ and $Y$ axes of object-labelled pixels.

Given an image $x_i$ of size $w \times w$ with the object bounding box represented in terms of minimum/maximum $XY$ coordinates as $(p_{min}^X, p_{max}^X, p_{min}^Y, p_{max}^Y)$, relative object size of the image $s_{x_i}$ is the ratio between the area of object bounding box and the total image area which can be computed

as follows:

$$s_{x_i} = \frac{(p_{max}^X - p_{min}^X)(p_{max}^Y - p_{min}^Y)}{w^2}.$$ (4)

The value of $s_{x_i}$ will be within the range of $[0, 1]$. Example images with different values of $s_{x_i}$ are shown in Figure 9.

We use $s_{x_i}$ of individual image samples to control object size characteristic of a dataset. In section 5, the datasets with small objects (i.e., ImageNetS919-SM and CUB-SM), are obtained by thresholding $s_{x_i}$ of image samples such that that their values are not larger than 0.2. In section 5.3, multiple thresholds of $s_{x_i}$ are employed on the ImageNetS919 dataset in order to study behavior of our models on different object size conditions. These thresholds are distributed uniformly from 0.05 to 1.0 with the step size of 0.05. The number of samples in each of these object size conditions is presented in Figure 10.

## A.2 ADDITIONAL ZERO-SHOT TRANSFER RESULTS

From table 1, we presented zero-shot performance of GC-CLIP variations with different model configurations. In this section, we provide full version of the results including performance of ViT-L/14 and DataComp in Table 3.

## A.3 GUIDED CROPPING WITH SUPERVISED MODELS

In the main paper, we mainly focus on applying our Guided Cropping to zero-shot models, i.e., CLIP and CALIP. We argue that Guided Cropping can be helpful in this case as image encoders of these models are designed to be generic so that they potentially encode non-discriminative information of input images.

Concerning our Guided Cropping component alone, it is, in fact, orthogonal to supervision strategies. Theoretically, our Guided Cropping can be employed with supervised models as well. In this case, models can be supervisedly trained as normal but, during inference, their input images can be cropped with our Guided Cropping component before forwarding to the models. In this section, we study behaviors of Guided Cropping when it is integrated with few-shot and fully-supervised models.

### A.3.1 FEW-SHOT MODELS

In this section, we conduct an experiment based on few-shot models, Tip-Adapter and Tip-Adapter-F (Zhang et al., 2021), to learn classification on ImageNetS919-SM and CUB-SM datasets in few-shot (n-shots=16 in our experiment). Its performance without and with Guided Cropping ($\alpha = 0.2$ with no box augmentation) is shown in the table below. According to the table, our Guided Cropping generally improves performance of Tip-Adapter variations. This empirically demonstrates benefits of our Guided Cropping for few-shot models.

### A.3.2 FULLY-SUPERVISED MODELS

In this section, we study behaviors of Guided Cropping when it is integrated with pretrained supervised models. In this regard, we utilize ImageNet pretrained models with ViT-B/32, ViT-B/16 and ViT-L/16 backbones from timm (Wightman, 2019), a deep learning library. These models are evaluated on ImageNetS919 and ImageNetS919-SM datsets with/without Guided Cropping. The results are shown in Table 5.

According to the results, optimal performance generally achieves with models without Guided Cropping or with Guided Cropping using large margin ratio, i.e., 0.8, whose crops already cover large context regions. We can observe this behavior even in the case of small objects (ImageNetS919-SM). These results indicate that, for these fully-supervised models, unrelated contexts generally do not degrade classification performance. In contrast, these contexts even improve their performance. This observation is actually not new and has been discussed in shortcut learning literature (Geirhos et al., 2020) that supervisedly trained networks can take unintended visual cues (e.g., background, texture) as shortcuts to gain classification performance on in-distribution samples.

Table 3: Zero-shot classification accuracies from different datasets and model configurations.

| Model | Prompt | Guided Cropping | Box Aug. | Dataset | | | |
|---|---|---|---|---|---|---|---|
| | | | | ImageNetS919 | CUB | ImageNetS919-SM | CUB-SM |
| CLIP (ViT-B/32) | Category | - | - | 63.62 | 51.83 | 52.83 | 49.57 |
| | | - | Random Crop | 64.42 | 52.45 | 53.47 | 50.79 |
| | | ✓ | - | 63.61 | 52.40 | 55.18 | 51.44 |
| | | ✓ | Random Crop | 64.46 | **53.12** | **56.00** | 52.81 |
| | | ✓ | Multi-Margin | **64.66** | **53.12** | **56.00** | **53.09** |
| | Descriptions | - | - | 68.54 | 53.05 | 55.70 | 50.14 |
| | | - | Random Crop | 69.15 | 53.62 | 57.33 | 50.79 |
| | | ✓ | - | 68.59 | 54.07 | 58.61 | **53.38** |
| | | ✓ | Random Crop | 69.07 | 54.47 | 59.08 | 53.09 |
| | | ✓ | Multi-Margin | **69.62** | **54.56** | **60.07** | 52.95 |
| CLIP (ViT-B/16) | Category | - | - | 68.60 | 56.51 | 57.75 | 55.54 |
| | | - | Random Crop | 68.81 | 56.89 | 58.05 | 57.41 |
| | | ✓ | - | 68.06 | 56.09 | 58.65 | 55.97 |
| | | ✓ | Random Crop | 68.19 | 56.78 | 58.35 | 57.12 |
| | | ✓ | Multi-Margin | **68.94** | **57.30** | **59.81** | **57.63** |
| | Descriptions | - | - | 72.67 | 57.78 | 61.61 | 56.55 |
| | | - | Random Crop | 73.17 | 58.87 | 62.13 | 57.99 |
| | | ✓ | - | 72.61 | 58.70 | 63.28 | **59.35** |
| | | ✓ | Random Crop | 72.86 | 58.99 | 63.32 | 58.78 |
| | | ✓ | Multi-Margin | **73.49** | **59.34** | **64.05** | 59.06 |
| CLIP (ViT-L/14) | Category | - | - | 75.15 | 63.08 | 64.78 | 62.16 |
| | | - | Random Crop | 75.30 | 63.32 | 64.70 | 62.59 |
| | | ✓ | - | 75.00 | 62.96 | 66.02 | 62.16 |
| | | ✓ | Random Crop | 75.04 | 63.24 | 66.54 | 62.73 |
| | | ✓ | Multi-Margin | **75.71** | **63.63** | **66.92** | **63.17** |
| | Descriptions | - | - | 78.48 | 64.65 | 67.78 | 63.17 |
| | | - | Random Crop | 78.65 | 64.60 | 67.65 | **63.96** |
| | | ✓ | - | 78.32 | 64.67 | 69.07 | 63.31 |
| | | ✓ | Random Crop | 78.28 | **64.88** | 69.41 | **63.96** |
| | | ✓ | Multi-Margin | **79.06** | 64.76 | **69.88** | 62.95 |
| DataComp (ViT-L/14) | Category | - | - | 82.05 | 85.57 | 69.88 | 85.18 |
| | | - | Random Crop | 82.10 | 86.07 | 69.84 | 86.04 |
| | | ✓ | - | 81.87 | 85.85 | 71.04 | 86.26 |
| | | ✓ | Random Crop | 81.75 | 85.99 | 71.04 | 86.04 |
| | | ✓ | Multi-Margin | **82.36** | **86.19** | **71.51** | **86.62** |
| | Descriptions | - | - | 82.66 | 86.04 | 70.01 | 86.12 |
| | | - | Random Crop | 82.82 | 86.45 | 70.48 | 86.98 |
| | | ✓ | - | 82.33 | 86.57 | 71.25 | 87.19 |
| | | ✓ | Random Crop | 82.23 | 86.62 | 71.25 | 87.19 |
| | | ✓ | Multi-Margin | **82.93** | **86.83** | **71.68** | **87.41** |

Comparing to cases of other supervision strategies, zero-shot and few-shot models are less likely to be affected by shortcut learning since exposing to none (or few) of samples on target datasets make them less likely to learn unintended visual clues from dataset biases.

## A.4 LOGIT REFINEMENT ON TOP-K PREDICTIONS

As per our method mentioned in section 4.1, after computing preliminary logits from conventional CLIP, only top-k predictions are considered and refined with Guided Cropping. We choose $k = 5$ in this work. In this section, we will provide reasons why we adopt this top-k refinement strategy. Two main reasons are given below.

Table 4: Few-shot performance with Tip-Adapter variations. Accuracies gain from Guided Cropping integration are given in parentheses.

| Model | Approach | Guided Cropping | Dataset ImageNetS919-SM | CUB-SM |
|---|---|---|---|---|
| ViT-B/32 | Tip-Adapter | - | 56.34 | 53.45 |
| | Tip-Adapter | ✓ | 58.27 (+1.93) | 54.53 (+1.08) |
| | Tip-Adapter-F | - | 62.43 | 60.22 |
| | Tip-Adapter-F | ✓ | 63.15 (+0.72) | 60.07 (-0.15) |
| ViT-B/16 | Tip-Adapter | - | 62.34 | 61.44 |
| | Tip-Adapter | ✓ | 64.05 (+1.71) | 62.30 (+0.86) |
| | Tip-Adapter-F | - | 68.04 | 67.12 |
| | Tip-Adapter-F | ✓ | 68.42 (+0.38) | 67.05 (-0.07) |
| ViT-L/14 | Tip-Adapter | - | 68.77 | 70.72 |
| | Tip-Adapter | ✓ | 70.44 (+1.67) | 71.94 (+1.22) |
| | Tip-Adapter-F | - | 72.24 | 73.88 |
| | Tip-Adapter-F | ✓ | 72.15 (-0.09) | 74.32 (+0.44) |

Table 5: Classification accuracies of ImageNet pretrained models with/without Guided Cropping on ImageNet919.

| Architecture | Guided Cropping | Margin Ratio | Box Aug. | Dataset ImageNetS919 | ImageNetS919-SM |
|---|---|---|---|---|---|
| ViT-B/32 | - | - | - | 76.82 | 61.53 |
| ViT-B/32 | - | - | Random Crop | 77.71 | 62.21 |
| ViT-B/32 | ✓ | 0.2 | - | 77.11 | 64.05 |
| ViT-B/32 | ✓ | 0.2 | Random Crop | 77.99 | **65.04** |
| ViT-B/32 | ✓ | 0.8 | - | 76.91 | 62.81 |
| ViT-B/32 | ✓ | 0.8 | Random Crop | **78.14** | 63.84 |
| ViT-B/16 | - | - | - | 81.72 | 68.89 |
| ViT-B/16 | - | - | Random Crop | **82.11** | **69.37** |
| ViT-B/16 | ✓ | 0.2 | - | 81.08 | 68.42 |
| ViT-B/16 | ✓ | 0.2 | Random Crop | 81.16 | 68.85 |
| ViT-B/16 | ✓ | 0.8 | - | 81.63 | 68.51 |
| ViT-B/16 | ✓ | 0.8 | Random Crop | 81.94 | **69.37** |
| ViT-L/16 | - | - | - | 86.09 | 75.62 |
| ViT-L/16 | - | - | Random Crop | 86.35 | **76.35** |
| ViT-L/16 | ✓ | 0.2 | - | 85.67 | 75.92 |
| ViT-L/16 | ✓ | 0.2 | Random Crop | 85.69 | 75.54 |
| ViT-L/16 | ✓ | 0.8 | - | 86.21 | 76.26 |
| ViT-L/16 | ✓ | 0.8 | Random Crop | **86.37** | **76.35** |

- Potential Accuracy: We found that there is already high chances that the correct classes are among predicted top-5 classes. To demonstrate this, we analyze top-1, top-5 and top-10 accuracies of conventional CLIP in Table 6. According to the results, large accuracy gaps can be noticed between top-1 and top-5 accuracies (24.53% for ImageNetS919 and 31.79% for CUB). In other words, by considering only 5 classes for refinement with Guided Cropping, upper bounds of final accuracies are already high. It must be noted that, while this upper bound accuracies can be raised further by considering top-10 classes, the gains compared to top-5 classes are relatively small. This may not worth introducing additional computation to the pipeline. Therefore, we decide to perform Guided Cropping based on predicted top-5 classes in this work.

- Common Bounding Boxes: We notice that visual appearances of top-5 classes are relatively similar in most cases. OWL-ViT is also likely to produce similar boxes for these classes. This makes the use of common bounding boxes (e.g., the primary box $b_i^0$ or the $\alpha$-margin box $b_i^\alpha$) among these classes reasonable. To illustrate this, considering each sample in

Table 6: Top-k accuracies from conventional CLIP (ViT-B/32) with category prompts.

| Dataset | Accuracy | | |
|---|---|---|---|
| | Top-1 | Top-5 | Top-10 |
| ImageNetS919 | 63.62 | 88.15 | 92.98 |
| CUB | 51.83 | 83.62 | 90.63 |

Figure 13 and 14, its primary box generally contains visual features which are (partially) similar to each top class making the box become a decent box candidate for all top classes.

## A.5 ACCURACIES WITH DIFFERENT OBJECT SIZE CONDITIONS

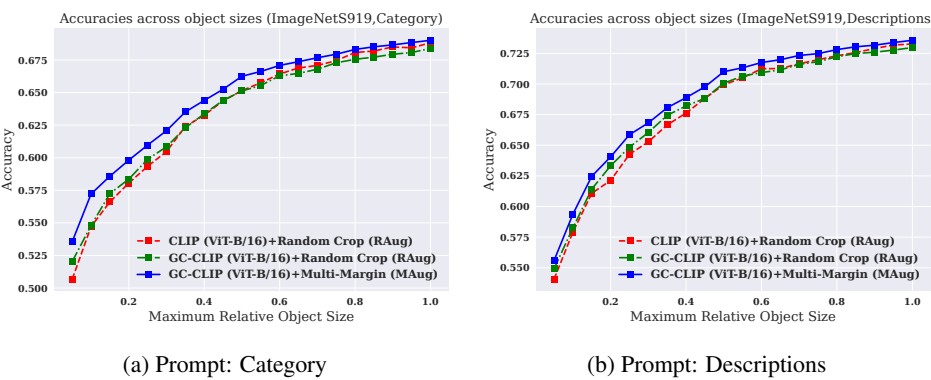

(a) Prompt: Category      (b) Prompt: Descriptions

Figure 11: Accuracies (ViT-B/16) on subsets of ImageNetS919 with various object size conditions.

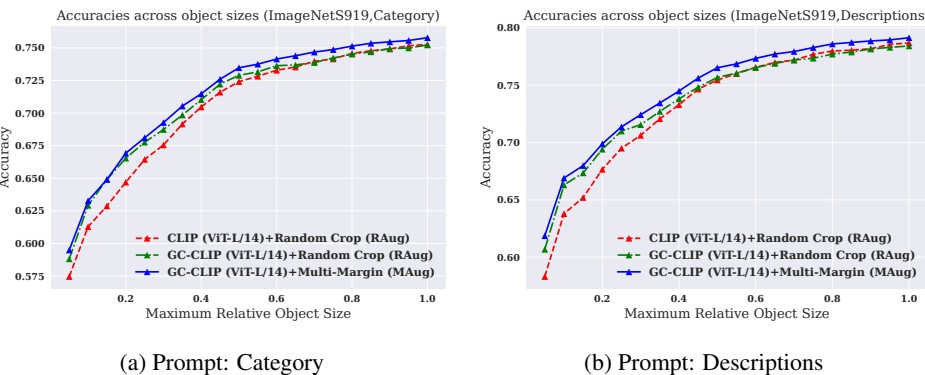

(a) Prompt: Category      (b) Prompt: Descriptions

Figure 12: Accuracies (ViT-L/14) on subsets of ImageNetS919 with various object size conditions.

In section 5.3, we study GC-CLIP performance on various object size conditions and show that GC-CLIP variations outperform baselines especially when target object sizes are small. The plots in Figure 6 are provided for models with ViT-B/32 backbone. In this section, additional evidences with other backbones are provided to support our claim. Figure 11 and 12 show similar plots for models with ViT-B/16 and ViT-L/14 backbones respectively. According to the figures, similar behavior can be observed. There are accuracy gaps between conventional CLIP and GC-CLIP and the gaps are larger on datasets with small objects. This demonstrates that our claim is consistent across different CLIP backbones.

## A.6 INFERENCE WITH OWL-VIT

OWL-ViT performs object detection taking images and text prompts as inputs and producing bounding boxes as well as their scores and class labels as outputs. In this work, for each image sample

Table 7: Accuracies from GC-CLIP (ViT-B/32) with different OWL-ViT inference strategies.

| Dataset | Prompt Type | Box Aug. | OWL-ViT Inference Single-Pass | Multi-Pass |
|---|---|---|---|---|
| ImageNetS919-SM | Category | RAug | 54.71 | **56.00** |
| ImageNetS919-SM | Category | MAug | 55.61 | **56.00** |
| ImageNetS919-SM | Descriptions | RAug | 57.84 | **59.08** |
| ImageNetS919-SM | Descriptions | MAug | 59.47 | **60.07** |
| CUB-SM | Category | RAug | 50.22 | **52.81** |
| CUB-SM | Category | MAug | **53.09** | **53.09** |
| CUB-SM | Descriptions | RAug | 51.51 | **53.09** |
| CUB-SM | Descriptions | MAug | **53.45** | 52.95 |

Table 8: Average similarity scores between images and their corresponding prompts (i.e., maximum logit values) of correctly classified samples of CLIP (with RAug) and GC-CLIP (with MAug) using ViT-B/32 backbone.

| Dataset | Prompt Type | Accuracy with CLIP | GC-CLIP |
|---|---|---|---|
| ImageNetS919-SM | Category | 29.39 | **29.71** |
| ImageNetS919-SM | Descriptions | 30.17 | **30.51** |
| CUB-SM | Category | 33.71 | **33.89** |
| CUB-SM | Descriptions | 34.30 | **34.55** |

$x_i$, we use OWL-ViT to extract bounding box candidates $B_i$ based on a set of detection prompts of the top-k classes $\{p_j^{det}|j \in J_i^k\}$. Theoretically, there are two possible options to obtain $B_i$ from OWL-ViT.

- Single Forward Pass (Single-Pass): with this option, an input image and all detection prompts are forwarded to OWL-ViT at once. With a single forward pass, OWL-ViT will produce a set of bounding boxes which will be used directly as $B_i$.

- Multiple Forward Passes (Multi-Pass): with this option, OWL-ViT will perform forward pass with one detection prompt at a time. In other words, there will be $k$ forward passes in total. Each forward pass will produce a set of bounding boxes $b_{ij}$ based on a detection prompt $p_j^{det}$. Bounding boxes estimated from all forward passes will be merged to get $B_i$ according to equation 2.

As mentioned in section 4.1, we decide to adopt Multi-Pass in our Guided Cropping pipeline as Multi-Pass is more robust to misdetection (if one pass fails, other passes can act as backup passes). In this section, we demonstrate empirically that Multi-Pass can lead to better performance.

In this regard, we conduct an experiment to compare GC-CLIP accuracies when Single-Pass and Multi-Pass are employed. The results are shown in Table 7. According to the results, GC-CLIP with Multi-Pass is consistently better across datasets and model configurations. This confirms our design choice to use Multi-Pass in our Guided Cropping pipeline.

### A.7 SIMILARITY BETWEEN CROPPED IMAGES AND THEIR PROMPTS

One motivation of our Guided Cropping is that, by minimizing unrelated information, CLIP image encoder can focus more on target objects leading to better image representations. In section 5.1 better image representations can be indirectly inferred via the improvement of the classification performance. In this section, we would like to analyze image representations in another perspective.

We argue that, if image representations are better, the representations should be not only less similar to prompts of other classes but also more similar to prompts of their own classes. In this regard, we investigate similarities of image embeddings (of the correctly classified samples) to their own prompts. Here, similarity scores are obtained in terms of maximum predicted logit values. Similarity score results of CLIP and GC-CLIP are shown in Table 8. We can notice that similarity scores

Table 9: Performance of GC-CLIP (ViT-B/32) on additional datasets using category-based prompts.

| Guided Cropping | Box Aug. | Dataset | | | | |
|---|---|---|---|---|---|---|
| | | ImageNet | ImageNetV2 | Stanford Dogs | ImageNet-A | ImageNet-R |
| - | - | 58.79 | 51.88 | 52.46 | 29.37 | 65.26 |
| - | Random Crop | 59.31 | 52.21 | 53.43 | 29.28 | 66.24 |
| ✓ | - | 58.95 | 52.84 | 53.92 | 31.41 | 65.47 |
| ✓ | Random Crop | 59.46 | 52.94 | **54.73** | 31.81 | 65.99 |
| ✓ | Multi-Margin | **59.84** | **53.30** | 54.12 | **31.97** | **66.67** |

between images and their corresponding prompts in case of GC-CLIP are consistently higher. This indicates that image representations after Guided Cropping are more similar to their prompts according to our assumption.

## A.8 VISUALIZING EXAMPLE RESULTS

In this section, we present top-5 logits estimated from CLIP and GC-CLIP on example samples from ImageNetS919 to demonstrate qualitatively that GC-CLIP can refine logits to make correct predictions. The results are illustrated in Figure 13 and 14.

## A.9 RESULTS ON ADDITIONAL DATASETS

In section 5, we aim to study the cases when objects of interest cover small areas of input images. Therefore, image classification datasets with segmentation/bounding box annotations are chosen for evaluation that enable us to quantify the performance on objects covering small areas. Hence, we choose ImageNetS919 and CUB for our evaluation as these datasets provide segmentation/bounding box annotations from which object sizes of image samples can be obtained. These annotations enable more insight studies with different object sizes. These datasets are also commonly used in weakly supervised object localization task (Zhu et al., 2022) as it needs similar annotations during evaluation.

For completeness, we perform evaluation on additional classification datasets without object size annotations as well. However, it must be noted that we may not be able to decouple effects of object size and extraneous image regions in this case. In this section, we present performance of GC-CLIP on ImageNet (Russakovsky et al., 2015), ImageNetV2 (Recht et al., 2019), Stanford Dogs (Khosla et al., 2011), ImageNet-A (Hendrycks et al., 2021b) and ImageNet-R (Hendrycks et al., 2021a) datasets.

The results are shown in Table 9. According to the results, even object sizes of these datasets are not controlled, our GC-CLIP is generally still better than the baselines. The magnitudes of improvement are generally similar to results in Table 1 in the main paper (refering unconstrained variants of ImageNetS919 and CUB).

One interesting observation which must be noted here is GC-CLIP performance on out-of-distribution datasets (i.e., ImageNet-A and ImageNet-R). We can observe that amounts of accuracy gains from GC-CLIP are different depending on out-of-distribution conditions. GC-CLIP benefits better on natural adversarial condition (ImageNet-A) than on rendition condition (ImageNet-R). We attribute this behavior to our dependency of OWL-ViT. In the rendition condition, objects are in unusual contexts such that OWL-ViT performance is not always consistent.

## A.10 COMPARISON WITH CENTRAL CROP

In our work, we demonstrate that image cropping guided by object locations can improve classification performance. To further support this argument, we perform experiments comparing our guided cropping with a deterministic cropping strategy, Central Crop, commonly used for classification (Jia et al., 2021; Zhai et al., 2022; Touvron et al., 2019).

Central Crop benefits under the assumption that target objects likely to locate at the center of input images. During inference, an input image will be cropped around its center according to a predefined

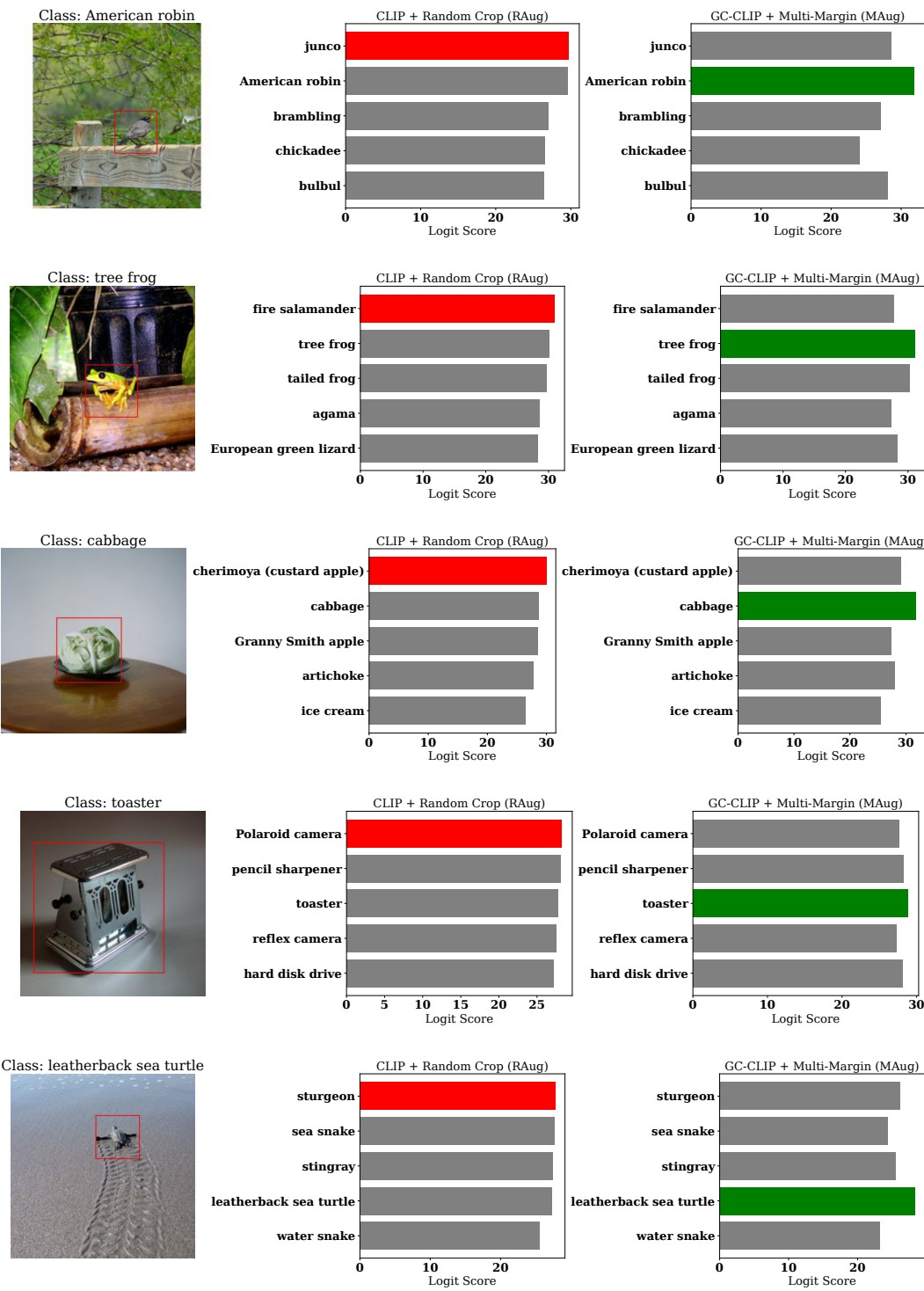

Figure 13: Top-5 logits on example samples improved by Guided Cropping (set 1). Model configurations are CLIP (with RAug) and GC-CLIP (with MAug) using ViT-B/32 backbone and prompt type of descriptions. Red boxes represent primary boxes used in our GC-CLIP pipeline.

cropping ratio from 0.0 to 1.0. The crop ratio of 1.0 refers to the usage of the full images without cropping. Then, the processed image will be resized to a compatible size for employed models be-

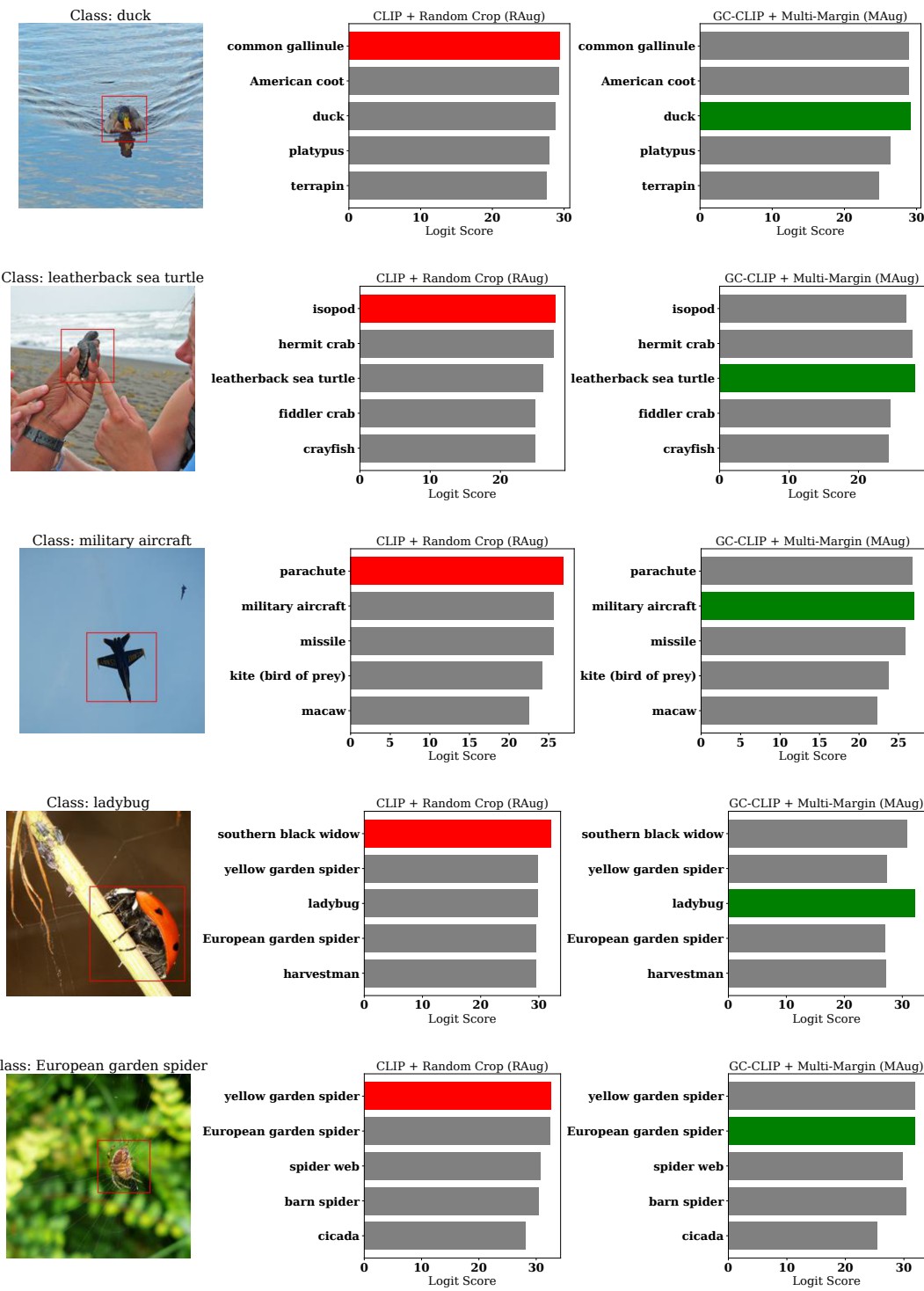

Figure 14: Top-5 logits on example samples improved by Guided Cropping (set 2). Model configurations are CLIP (with RAug) and GC-CLIP (with MAug) using ViT-B/32 backbone and prompt type of descriptions. Red boxes represent primary boxes used in our GC-CLIP pipeline.

fore performing the inference. We conduct experiments with Central Crop using different cropping ratios on ImageNetS919-SM. Its performance can be visualized as in Figure 15.

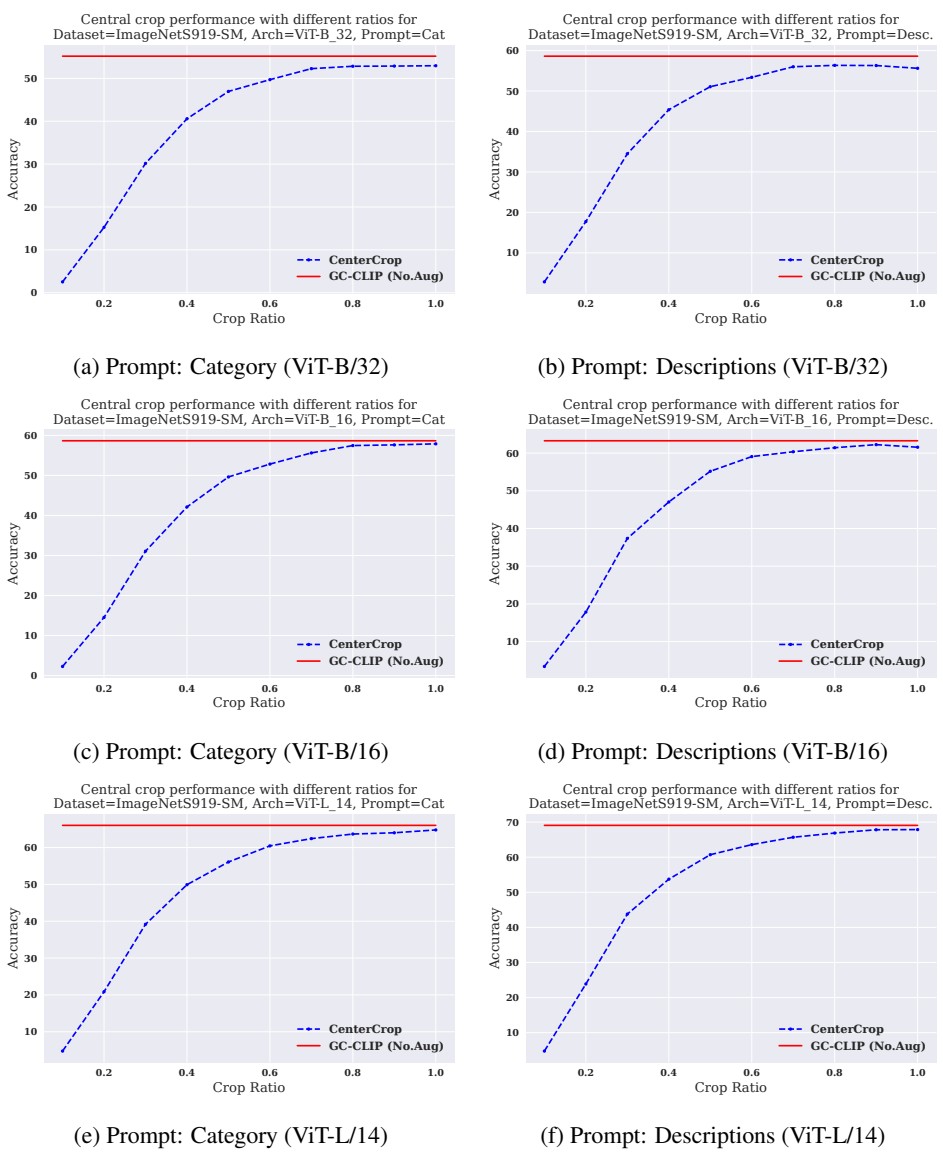

(a) Prompt: Category (ViT-B/32)  (b) Prompt: Descriptions (ViT-B/32)

(c) Prompt: Category (ViT-B/16)  (d) Prompt: Descriptions (ViT-B/16)

(e) Prompt: Category (ViT-L/14)  (f) Prompt: Descriptions (ViT-L/14)

Figure 15: Central crop performance with different cropping ratios compared to GC-CLIP (without box augmentation) on ImageNetS919-SM.

According to the results, we can see that, models with Central Crop can slightly improve performance compared to vanilla models. For example, according to Figure 15b, the model without Central Crop (ratio=1.0) achieves the accuracy of 55.61 while the model with Central Crop (ratio=0.9) achieves the higher accuracy of 56.30. However, on Figure 15, models with Guided Cropping (without box augmentation) consistently outperform Central Crop. This supports the argument that our cropping approach guided by object locations is preferable over simple cropping at a predefined location.

