# OpenReview forum: "Zero-Shot Visual Classification with Guided Cropping"
_ICLR.cc/2024/Conference — Submitted to ICLR 2024_

### Official Review · Reviewer_KVmG · 2023-10-30

**Soundness:** 2 fair
**Presentation:** 3 good
**Contribution:** 2 fair
**Rating:** 6
**Confidence:** 4

**Summary:**

This paper proposes GC-CLIP to improve the zero-shot transfer performance for image classification tasks at inference time. It uses the existing open vocabulary detector (OWL-ViT) to select the bounding box of the main object, and proposes Multi-Margin Box Augmentation (MAug) to avoid losing potentially useful context information.

The result is positive but not significant on ImageNetS919 and CUB classification benchmarks, in which the gap is larger on datasets particularly for small objects.

**Strengths:**

The method is simple and shows positive results.

Ablations are comprehensive and solid.

Both qualitative and quantitative results are supplied.

Well written and easy to read.

**Weaknesses:**

ImageNetS919 and CUB are carefully selected as the benchmarks, but the number of the benchmarks are usually too limited for zero-shot classification evaluations. For more solid results, it would be nice to also report results on common image classification tasks, such as ImageNet, VTAB, and OOD benchmarks (e.g. ObjectNet) etc.

The cost of this method was not clearly mentioned, which is meaningful given an additional detector is needed during the inference time.

In Table 2, the comparison between CALIP and GC-CLIP is not solid: we may be able to also say “CALIP can improve performance over vanilla GC-CLIP” (if this is the truth). The result in Table 2 is also hard to compare with Table 1. So I would suggest reusing the same setup as Table 1 to compare CALIP and GC-CLIP for a clearer result.

Even though the empirical results are positive, the improvement looks a bit marginal, which might be less insightful to the community.

**Questions:**

as mentioned in “Weaknesses”:
- could we report more results from more common image classification benchmarks?
- could we report the additional inference cost?

mild comments:
- In 5.4, could we also show the scores of the examples? So that we could understand a bit more possibly the confidence of the model (e.g. whether a smaller score could indicate that the model needs more context for classification?).
- In 5.2: “Too tight bounding boxes can make the models have unclear information…” instead of “...having…”
- In 5.4: the format of quotes seems wrong: ”land” and ”sea”

---

> ### Author Response · Authors · 2023-11-18
>
> Thank you for your valuable comments as well as suggestions. We address your questions and concerns as follows.
>
> > could we report more results from more common image classification benchmarks?
>
> **Response:** Thank you for your suggestion. ImageNetS919 and CUB datasets are chosen in our work as these datasets are classification datasets with annotations (e.g., bounding boxes) from which object sizes can be retrieved. This allows us to perform detailed analysis under different object size conditions. These datasets are also commonly used in weakly supervised object localization task (e.g., [1]) as it needs similar annotations during evaluation.
>
> Nevertheless, for the completeness of our paper, as suggested by the reviewer, we perform additional evaluation on more common datasets (ImageNet, ImageNetV2, Stanford Dogs). The results can be found in the appendix A.9 in the updated version of our paper. The results still demonstrate that integration of Guided Cropping can still be beneficial even in datasets with unconstrained object sizes (similar to the unconstrained variants of ImageNetS919 and CUB).
>
> [1] "Weakly Supervised Object Localization as Domain Adaption", CVPR, 2022.
>
> > could we report the additional inference cost?
>
> **Response:** Additional parameters are required for OWL-ViT that attributes to a total of 153M model parameters. When performing a single sample inference, additional inference time required on top of CLIP is 0.15 and 0.16 seconds without and with box augmentation on a single Tesla V100-SXM2 GPU respectively. We will include these details in the final version of the paper.
>
> > In Table 2, the comparison between CALIP and GC-CLIP is not solid: we may be able to also say “CALIP can improve performance over vanilla GC-CLIP” (if this is the truth). The result in Table 2 is also hard to compare with Table 1. So I would suggest reusing the same setup as Table 1 to compare CALIP and GC-CLIP for a clearer result.
>
> **Response:** Thank you for your constructive feedback. We have adapted Table 2 in the updated version of our paper to have the same setup as Table 1 accordingly so that both tables are now comparable. We can see that, even without box augmentation, vanilla GC-CLIP gives better performance compared to CALIP. Additionally, Guided Cropping can also be integrated into CALIP to further improve the performance. We have included this discussion in the updated version of our paper.
>
> > In 5.4, could we also show the scores of the examples? So that we could understand a bit more possibly the confidence of the model (e.g. whether a smaller score could indicate that the model needs more context for classification?).
>
> **Response:** Thank you for your suggestion. Visualizing the scores of these examples can indeed improve understanding of the model behaviors. We will incorporate this suggestion in the final version of our paper.

---

> > ### Comment · Reviewer_KVmG · 2023-11-20
> >
> > Thanks for all the updates!
> >
> > > This allows us to perform detailed analysis under different object size conditions. These datasets are also commonly used in weakly supervised object localization task (e.g., [1]) as it needs similar annotations during evaluation.
> >
> > I agree that the datasets with box ground-truth is necessary for analysis, but given the proposed method is to improve zero-shot classification capability, to make the result more convincing, besides analysis, showing the results on common classification tasks is also necessary.
> >
> > > Nevertheless, for the completeness of our paper, as suggested by the reviewer, we perform additional evaluation on more common datasets (ImageNet, ImageNetV2, Stanford Dogs). The results can be found in the appendix A.9 in the updated version of our paper.
> >
> > Thanks for the additional evaluations! A bit more comments:
> > 1. for baselines, central crop is a common way in zero-shot evals [1], as it often shows better results than resize-only etc., so it would be more convincing if we could compare the proposed method with central crop evals as well;
> > 1. for completeness, to convince others that the proposed method is generally better for classification, we might need to add more classification tasks, such as VTAB, and OOD tasks (ObjectNet, Imagenet-A etc.).
> > 1. IIUC the proposed method is a general way to improve zero-shot evals, so it would be more complete if we add zero-shot retrieval results (e.g. COCO, Flickr) as well for completeness and allowing more insights.
> >
> > [1] https://arxiv.org/abs/2102.05918, https://arxiv.org/abs/2111.07991

---

> > > ### Author Response · Authors · 2023-11-22
> > >
> > > > for baselines, central crop is a common way in zero-shot evals [1], as it often shows better results than resize-only etc., so it would be more convincing if we could compare the proposed method with central crop evals as well;
> > >
> > > **Response:** Thank you for suggesting an relevant baseline. We agree that comparing our approach against Central Crop would be helpful to see the difference between static cropping strategy (Central Crop focuses only on center regions) and dynamic cropping strategy (Guided Cropping focuses on regions where target objects locate). In this regard, we provide results of this additional experiment in appendix A.10 in the updated version of the paper. In the experiment, we sweep cropping ratio of Central Crop from 0.1 to 1.0. We can observe that models with Central Crop provides better performance than vanilla classifiers in some cases. However, Guided Cropping still outperforms Central Crop. This demonstrates that cropping strategy focusing on objects is preferable.
> > >
> > > > for completeness, to convince others that the proposed method is generally better for classification, we might need to add more classification tasks, such as VTAB, and OOD tasks (ObjectNet, Imagenet-A etc.).
> > >
> > > **Response:** Thank you for your suggestion. Conducting experiments on OOD data indeed allows better understanding of GC-CLIP. We conduct additional experiments on ImageNet-A and ImageNet-R (we would apologize not to be able to provide result for ObjectNet during this rebuttal due to technical reason). We have included these results in the updated version of our paper (appendix A.9). According to the results, amounts of accuracy gains from GC-CLIP are different depending on out-of-distribution conditions. GC-CLIP benefits better on natural adversarial condition (ImageNet-A) than on rendition condition (ImageNet-R). We attribute this behavior to our dependency of OWL-ViT. In the rendition condition, objects are in unusual contexts such that OWL-ViT performance is not always consistent.
> > >
> > > > IIUC the proposed method is a general way to improve zero-shot evals, so it would be more complete if we add zero-shot retrieval results (e.g. COCO, Flickr) as well for completeness and allowing more insights.
> > >
> > > **Response:** Our Guided Cropping aims to produce better image representations based on known target classes of interest. It is designed for classification task in which all target classes are given. In case of image-text retrieval, possible target output space is large and not completely defined. Therefore, zero-shot retrieval is not a compatible benchmark in our case (we will address this limitation in the final version of the paper). One could try using Guided Cropping to refine representations of candidate images in text-to-image retrieval but this would not be straightforward. We would leave this extension as a potential future work.

---

> > > > ### Comment · Reviewer_KVmG · 2023-11-22
> > > >
> > > > Thanks for the updates! Most of my concerns have been addressed. However, I would encourage the authors to continue improving the paper's completeness in the final version, including:
> > > >
> > > > 1. Explain why in Stanford Dogs GC-CLIP doesn't work: because of OWL-ViT or lack of context etc.?
> > > >
> > > > 1. Add more common and OOD classification datasets to make the conclusion more convincing. The current "ImageNet, ImageNetV2, Stanford Dogs" and "ImageNet-A and ImageNet-R" are still too few compared to other relevant literatures.
> > > >
> > > > 1. Add [ImageNet-Hard](https://arxiv.org/abs/2304.05538) as another baseline, which proposes "a test-time augmentation (TTA) technique that improves classification accuracy by forcing models to explicitly perform zoom-in operations before making predictions", so should be very relevant.

---

> > > > > ### Author Response · Authors · 2023-11-23
> > > > >
> > > > > We are pleased to know that most of the reviewer's concerns are addressed. We thank the reviewer for considering our rebuttal and increasing the score towards acceptance.
> > > > >
> > > > > Regarding to the evaluation on stanford dogs, according to the results in Table 9, models integrated with Guided Cropping are still better than the ones without Guided Cropping. In other words, Guided Cropping still works. However, MAug may not be necessary useful in this domain. As target objects are dogs, context information outside object boundaries is generally non-discriminative. Consequently, integration with MAug is not necessary better than RAug. We would definitely include this discussion in the final version of our paper. Additionally, we will definitely include additional datasets and baselines as suggested by the reviewer.

---

### Official Review · Reviewer_tiUR · 2023-10-31

**Soundness:** 3 good
**Presentation:** 3 good
**Contribution:** 2 fair
**Rating:** 5
**Confidence:** 3

**Summary:**

The paper proposes an inference pipeline that combines SOTA zero-shot image classification models (e.g. CLIP) with SOTA open vocabulary object detection models (e.g. OWL-ViT) to improve zero-shot classification of images with smaller objects. Notably, the proposed approach does not require any model training. The model relies on a combination of CLIP for whole-image and image crop classification and OWL-ViT for object localisation and cropping. The paper explores several hyper-parameters of the proposed pipeline (e.g. margin of the crops and random crop augmentation) and demonstrates that their method improves on several CLIP-based baselines and on OWL-ViT.

**Strengths:**

* Solid paper, clearly written and well-motived.
* The method is pragmatic, and seemingly driven by practical considerations of actually using CLIP and OWL-ViT models in real life applications.
* The proposed inference pipeline does not require any training and thus can be readily used for many applications.

**Weaknesses:**

* The observations of current limitations of CLIP and OWL-ViT models are somewhat surface-level and I believe well-known (although possibly not written down in a publication)
* The proposed solution to the observed limitations (i.e. the proposed inference pipeline) is as far as I know novel, but maybe better presented at a more computer vision focused conference.
* Although the focus of the proposed approach is to correct failure cases of CLIP and OWL-ViT (i.e. small object classification), and the benchmarks were chosen to assess this specifically, it would be very useful to see the proposed method benchmarked on widely used zero-shot datasets like ImageNet.
* It would be interesting to discuss the limitations of the proposed method. For example, it focuses on image classification, but CLIP and OWL-ViT provide more than that. For example, CLIP embeddings can be used for image-text retrieval, and OWL-ViT embeddings can be used for image-conditioned detection. Can the authors' method be extended to these use cases?

**Questions:**

See Weaknesses above.

---

> ### Author Response · Authors · 2023-11-18
>
> Thank you for your valuable comments as well as suggestions. We address your questions and concerns as follows.
>
> > The observations of current limitations of CLIP and OWL-ViT models are somewhat surface-level and I believe well-known (although possibly not written down in a publication)
>
> **Response:** In our work, we address two limitations of CLIP, its vulnerability to extraneous image regions and its sensitivity to non-semantic crops. I agree with the reviewer that some can have assumptions similar to these limitations. However, validating the assumptions and systematically quantifying impacts of these limitations are also crucial but missing from the community. For the vulnerability to extraneous image regions, we show that performance gaps between scenarios with unconstrained objects and scenarios with small objects can be up to 15-20 percents (see Figure 6 in section 5.3). The performance drops can also be observed in various supervision strategies, i.e., in zero-shot, few-shot and fully supervised models (see Table 3, 4 and 5 in the appendix). For the sensitivity to non-semantic crops, its effect is quantitatively evaluated. From Figure 4, we demonstrate that around 40% of test samples in our setup have changes in their final predictions just from small removal around image boders. These empirical studies show that impacts of these limitations are not negligible. We believe that these studies can be helpful for the research community.
>
> > Although the focus of the proposed approach is to correct failure cases of CLIP and OWL-ViT (i.e. small object classification), and the benchmarks were chosen to assess this specifically, it would be very useful to see the proposed method benchmarked on widely used zero-shot datasets like ImageNet.
>
> **Response:** Thank you for your suggestion. We conducted further experiments on additional datasets (ImageNet, ImageNetV2 and Stanford Dogs). The results can be found in the appendix A.9 of the updated version of our paper. The results still demonstrate that GC-CLIP improves baseline performance even in datasets with unconstrained object sizes.
>
> > It would be interesting to discuss the limitations of the proposed method.
>
> **Response:** Guided Cropping can be viewed as a strategy to refine image features conditioned to classes of interest. Therefore, it is ideal to be employed in the setting of classification task since all classes of interest are known in advance. For some generic tasks (image-text retrieval, image-conditioned detection) that all classes of interest are not defined, it would not be straigtforward to employ Guided Cropping in general.
>
> However, on the specific scenarios that domains of interest are known in advance, Guided Cropping can also be beneficial. For example, in case of image-conditioned detection, if we already know that the domain of interest is animal, we can use Guided Cropping to refine image features using generic animal prompts (e.g., the word "animal" itself). In this case, information of unrelated contexts of query images can be discarded by Guided Cropping which could lead to better detection performance. We will include this discussion in the final version of the paper.

---

### Official Review · Reviewer_EzG9 · 2023-11-01

**Soundness:** 2 fair
**Presentation:** 2 fair
**Contribution:** 1 poor
**Rating:** 3
**Confidence:** 5

**Summary:**

The paper proposes a method to crop input image to obtain more robust features for zero-shot object classification. The author use a pre-trained zero-shot object detection model to obtain an initial bounding box that is most responsive to an input prompt. The box is then enlarged before inputing into CLIP for classification. This approach brings consistent ZSC improvement using ViT-B model as baseline, tested on ImageNetS919 and CUB datasets.

**Strengths:**

The approach is simple and easy to re-implement.

**Weaknesses:**

The novelty is limited. Many papers [1,2,...] have discussed the impact of cropping in image classification. The paper aims to find an optimal crop but there is no technical contribution since the heavy-lifting is done purely based on the pre-trained object detector. Perhaps the core contribution is to show that an object detector can be used for this purpose? I think it is incremental.

The potential applicability is limited. The method is very specific to CLIP and the core method doesn't work right off the bat but requires post-processing steps on top of the initial boxes. Further, this method is only suitable for classifying images whose labels associating to a small object it contains. The authors did provide some qualitative evaluation on these cases but I think it could benefit the paper to frame it more aggressively into this direction since it seems to me this is the only scenario where it might prove a significant advantage.

[1] ImageNet-Hard: The Hardest Images Remaining from a Study of the Power of Zoom and Spatial Biases in Image Classification - Taesiri et al. NeurIPS 2023.
[2] Generating Features with Increased Crop-related Diversity for Few-Shot Object Detection - Xu et al. CVPR 23

**Questions:**

N/A

---

> ### Author Response · Authors · 2023-11-18
>
> Thank you for your valuable comments as well as suggestions. We address your questions and concerns as follows.
>
> > The novelty is limited. Many papers [1,2,...] have discussed the impact of cropping in image classification.
>
> **Response:** Thank you for your feedback. According to the mentioned papers, we agree that the paper [1] is closely related to ours. However, our work provides some additional insights which have not discussed in [1]. Firstly, while [1] provides the evidence that optimal crops can be helpful for classifiers, [1] has not discussed the properties which the optimal crops should have. In our work, we demonstrate that the optimal crops should be close to target objects but not too close. We empirically provide evidence to support this argument by varying multiple margin ratios as presented in section 5.2. Additionally, we show that, even though emphasizing on object information is preferable, we should not ignore the context information completely. Some strategies to balance object and context information must be developed. This is the main motivation for us to propose Multi-Margin box augmentation. Lastly, the aggregation-based approach proposed in [1] is also similar to our Random Cropping variants in the sense that they determine crops from static strategies without considering where target objects are. Our Guided Cropping takes locations of the target objects into account leading to better performance. We will include this discussion in the final version of our paper.
>
> > The potential applicability is limited. This method is only suitable for classifying images whose labels associating to a small object it contains. The authors did provide some qualitative evaluation on these cases but I think it could benefit the paper to frame it more aggressively into this direction since it seems to me this is the only scenario where it might prove a significant advantage.
>
> **Response:** We understand your concern. To strengthen our motivation, we would like to give an example of potential scenarios in which recognition of small objects is beneficial. One example scenario is under automated driving application. In this scenario, recognizing objects which are far away is crucial. It allows automated vehicles to have better decision making resulting in safer maneuver. As the *far away* objects are mostly small in image frames, our Guided Cropping can be employed in this case to improve recognition performance for those objects. We will include this motivation in the final version of the paper.
>
> > The method is very specific to CLIP and the core method doesn't work right off the bat but requires post-processing steps on top of the initial boxes.
>
> **Response:** Our Guided Cropping aims to improve models whose performances are likely to be degraded by extraneous, non-discriminative image regions. In our work, we show that CLIP is one of the models which has this mentioned behavior so that its integration with Guided Cropping can be beneficial. In principle, Guided Cropping can be applied not only to CLIP but also other models which are vulnerable to extraneous non-discriminative image regions. For example, we demonstrate that Guided Cropping integration with CALIP and Tip-Adapter can be helpful (see Table 2 and Table 4). Other potential relevant models are few-shot models as observed by [1] that they suffer from shortcut learning due to background information.
>
> Regarding to the requirement of post-processing steps on top of the initial boxes, we do not see this as a disanvantage of our work but a contribution. In fact, using the initial boxes directly can already improve performance. The post-processing only aims to exploit the availability of the boxes as much as possible.
>
> [1] "Rectifying the Shortcut Learning of Background for Few-Shot Learning", NeurIPS, 2021.

---

### Official Review · Reviewer_LrWg · 2023-11-03

**Soundness:** 3 good
**Presentation:** 4 excellent
**Contribution:** 2 fair
**Rating:** 5
**Confidence:** 4

**Summary:**

In this work, the authors identify that the Image Encoder of CLIP is more inclined to extract generic image representation, thus leading to performance degradation in zero-shot closed-set object classification tasks, especially for small objects. To address this problem, they proposed GC-CLIP, which crops and zooms in on the target itself by introducing a guided cropping method based on a zero-shot target detection model, thus improving the performance of CLIP.

**Strengths:**

1- The language is clearly presented. The authors use precise and concise language so that the reader can easily understand the background, methodology, and results of the study.
2- Ablation studies are comprehensive. The authors demonstrated the superiority of GC-CLIP over CLIP through many ablation studies and analysed various factors.

**Weaknesses:**

1- I suggest the authors report the computational cost of GC-CLIP in the paper, including the parameters, FLOPs or the inference time, for a more comprehensive comparison with CLIP.
2- I am confused about the necessity of combining OWL-ViT and CLIP, because the authors’ results in the experimental section show that the difference between introducing OWL-ViT for guided cropping and using random cropping is slight, more results and analysis on different datasets should be provided to illustrate the advancement of guided cropping. In particular, the authors did not report results with random cropping alone when guided cropping was used on CALIP.
3- An essential prerequisite for the successful application of OWL-ViT in the GC-CLIP is that OWL-ViT can provide a detection box for every target in the image. The authors have yet to carry out validation on more datasets to verify the impact and constraints of the detectors on their method, so it cannot judge the processing performance of GC-CLIP for other more complex datasets, and more analysis is needed.
4- Authors should report the performance comparison of GC-CLIP with current popular methods.
5- Authors also need to check for grammatical problems. For example, in the last sentence of paragraph 5 of the introduction section, there is a subject-verb inconsistency between “the cropped image” and “decrease” and “result in”.

**Questions:**

Please refer to Weaknesses.
My main concern is the necessity of this guided cropping approach as it seems to have less difference in performance compared to what random cropping brings. There is a need for more results on more datasets and comparisons with other popular methods to demonstrate the performance of GC-CLIP.

---

> ### Author Response · Authors · 2023-11-18
>
> Thank you for your valuable comments as well as suggestions. We address your questions and concerns as follows.
>
> > I suggest the authors report the computational cost of GC-CLIP in the paper.
>
> **Response:** Additional parameters are required for OWL-ViT that attributes to a total of 153M model parameters. When performing a single sample inference, additional inference time required on top of CLIP is 0.15 and 0.16 seconds without and with box augmentation on a single Tesla V100-SXM2 GPU respectively. We will include these details in the final version of the paper.
>
> > In the experimental section show that the difference between introducing OWL-ViT for guided cropping and using random cropping is slight.
>
> **Response:** We understand that your concern about slight improvement mainly refers to results on datasets with unconstrained object sizes (ImageNetS919 and CUB). In this case, this observation is expectable since many images of these datasets already have large enough object sizes (images have small extraneous regions) and thus do not greatly benefit from our Guided Cropping component. Note that our work is mainly motivated and designed to address the challenging cases of input images with small object sizes and large extraneous regions. Henceforth, our improvements are more evident on ImageNetS919-SM and CUB-SM (curated with objects with small size), where the improvements from random cropping can be high up to almost 3 percent.
>
> > The authors did not report results with random cropping alone when guided cropping was used on CALIP.
>
> **Response:** Thank you for your suggestion. We have integrated random cropping box augmentation with CALIP and updated results in Table 2 to be more detailed. The results still show the benefit of combining Guided Cropping with CALIP.
>
> > An essential prerequisite for the successful application of OWL-ViT in the GC-CLIP is that OWL-ViT can provide a detection box for every target in the image. The authors have yet to carry out validation on more datasets to verify the impact and constraints of the detectors on their method, so it cannot judge the processing performance of GC-CLIP for other more complex datasets, and more analysis is needed.
>
> **Response:** GC-CLIP can be viewed as a framework utilizing available object detectors to improve classification performance. In our work, we use OWL-ViT as a candidate of object detectors. We agree that the performance of OWL-ViT can be limited but we would like to show that it is good enough for improving classification performance under our framework. To demonstrate benefits of our setup in other scenarios, as suggested by the reviewer, we have conducted further experiments on additional common classification datasets (ImageNet, ImageNetV2 and Stanford Dogs) with their original test splits. The results are presented in the appendix A.9 in the updated version of our paper. According to the results, our Guided Cropping strategy can still consistently improve performance of the baselines.
>
> > Authors should report the performance comparison of GC-CLIP with current popular methods.
>
> **Response:** CALIP is actually one of the recent models from AAAI-2023 we consider in our work. Anyway, we also agree that including more baselines would make our results statistically more meaningful. In this regard, we incorporate a CLIP(ViT-L/14) variation trained with DataComp from NeurIPS-2023 [1] as one of our baselines. Its performance is reported in Table 3 of the appendix A.2 in the updated version of our paper. It must be noted that performance of DataComp can still be further improved by integrating with our Guided Cropping. This demonstrates consistent observations among various baselines.
>
> [1] "DATACOMP: In search of the next generation of multimodal datasets", NeurIPS, 2023.
>
> > Authors also need to check for grammatical problems.
>
> **Response:** Thank you for your comments. We have updated our text accordingly.

---

### Author Response · Authors · 2023-11-22

Dear reviewers and ACs,

We would like to thank all reviewers for valuable suggestions and feedback. We updated our paper according to our discussion with reviewers. The main changes from the original version can be summarized below.

* In appendix A.9, we evaluate GC-CLIP on additional classification datasets including common datasets (ImageNet, ImageNetV2, Stanford Dogs) as well as OOD datasets (ImageNet-A and ImageNet-R).
* In appendix A.10, we compare GC-CLIP with another relevant baseline, Central Crop. This demonstrates advantage of cropping dynamically based on object locations over cropping with predefined locations.
* In appendix A.11, we provide discussion regarding to limitation of GC-CLIP.
* In appendix A.2, we consider another recent CLIP variation trained with DataComp as another baseline. We also include details of additional computation costs required for GC-CLIP.
* For Table 2, we modify evaluation configurations of CALIP to be comparable with Table 1. This demonstrates advantage of GC-CLIP even without box augmentation.

We would be really grateful if the reviewers could revise their reviews based on our responses and the updated version of our paper. Additionally, we also welcome any additional comments or questions.

---

### Meta-Review · Area_Chair_FbLN · 2023-12-08

**Metareview:**

The paper proposes a guided cropping method for zero-shot closed-set object classification using CLIP. The model adopts OWL-VIT to localize a potential target (small) object in the input image and improve the CLIP classification accuracy with guided cropping.

The paper received 1 reject, 2 weak rejects, and 1 weak accept overall leaning towards rejection. The main concerns raised by the reviewers are about its limited significance in the novelty and effectiveness. Also, the concerns about the additional computational costs are not properly addressed in the rebuttal.

Overall, the flaws outweigh the advantages and therefore the AC recommends the rejection of the paper.

**Justification For Why Not Higher Score:**

The majority of the reviewers recommended the rejection and many significant concerns are not fully addressed in the rebuttal.

**Justification For Why Not Lower Score:**

N/A

---

### Decision · Program_Chairs · 2024-01-16

Reject